# Brothers in Arms? How Neoliberalism Connects North and South Higher Education: Finland and Portugal in Perspective

**Sara Margarida Diogo** [1,2,*] and **Teresa Carvalho** [1,2]

1 Department of Social, Political and Territorial Sciences, University of Aveiro, Campus Universitário de Santiago, 3810-193 Aveiro, Portugal; teresa.carvalho@ua.pt
2 CIPES—Research Center on Higher Education Policies, 4450-227 Matosinhos, Portugal
* Correspondence: sara.diogo@ua.pt

**Abstract:** This paper puts in perspective the reforms of the Portuguese and Finnish higher education (HE) sectors in the light of the role intergovernmental organisations have—especially the Organisation for Economic Cooperation and Development (OECD)—in influencing neoliberal public policies in these countries. On the year that the OECD celebrates its 62nd anniversary, (the OECD was founded with this name on 14 December 1960 by 20 countries, following the establishment of the former European Economic Co-operation (OEEC) in April 1948) and by comparing two different countries, this article analyses the extent to which the OECD has been and is an "imperial agent" in Portuguese and Finnish HE policies. By cross-comparing the OECD reports of both HE systems, the empirical data shows how the OECD proposes neoliberal reforms based on three main components of neoliberalism: market, management and performativity in different countries. Taking these proposals into account, Portugal and Finland undertook similar HE legislative reforms despite their geographical, historical, cultural and economic differences. The data reveal a convergence in HE policies in these countries, anticipating the reinforcement of neoliberal policies at the national level.

**Keywords:** OECD; Portugal; Finland; HE; policy diffusion; policy implementation; New Public Management (NPM)

## 1. Introduction

In the last few decades, public administration reforms in all the traditional main areas of welfare states (e.g., health, education and social security) have been a common activity of governments in post-industrial democracies, implemented with a considerable degree of similarity to the proposed changes and the legitimating discourses sustaining them (Pedersen 2006; Torres 2004; Peters 1997). These proposed changes and discourses are highly influenced by or aligned with neoliberal ideas (Watermeyer and Olssen 2016). The same holds true for higher education (HE) policies, with it being widely recognised that different national governments resort to neoliberal ideas to find solutions to facing (imposed) external challenges (Ball 2012, 2016; Amaral et al. 2003; Olssen and Peters 2005; Torres and Schugurensky 2002; Hill and Kumar 2012; Watermeyer and Olssen 2016).

The hegemony of discourses and practices associated with neoliberal reforms induce an acculturation process that results in "education policies with similar principles in countries in all continents, with very different cultural and political histories" (Ball 2016, p. 1046). In this context, neoliberalism can be classified as a "travelling ideology", since it has become central in political discourses both in consolidated HE systems and in emergent ones, as well as both in right and left governments (Hill and Kumar 2012).

The role of intergovernmental organisations as the supporting source of this "travel" has been identified in different domains, including in HE (Shahjahan 2012; Ball 2016; Martens and Wolf 2009; Amaral et al. 2003; Ball 2003; Davies 2001). However, it is still not clear how intergovernmental organisations (effectively) promote convergence in HE policies in different countries. Concerning this, the institutional perspective (Strang and Meyer

1993; Meyer and Rowan 1977; Meyer et al. 2008; Dolowitz and Marsh 2000; DiMaggio and Powell 1983; Diogo et al. 2015) and, more specifically, sociological institutionalism shed light on the mechanisms through which the convergence of policies, ideas and administrative arrangements between nations or fields of organisations is promoted by intergovernmental organisations. For example, when studying the process of "Europeanisation" in Southern European countries, Featherstone and George (2014) explicitly referred to the usefulness of the sociological institutionalism perspective in understanding the diffusion of new policy paradigms "from the core to the periphery" and the mechanisms or schemes inducing domestic structural transformations. In turn, authors such as Gonçalves and Papon (2004) also referred to the role of "Europeanisation" in the globalisation of research, drawing attention to the consequences this might entail (e.g., the reduction—and increasing asymmetries—of national funding support).

Portugal and Finland are two European countries with different historical, geographical, economic, cultural and structural features. In 2018, Finland occupied the 15th position in the ONU Human Development Report, while Portugal was ranked 41st (United Nations Development Programme (UNDP) 2015). Focusing on two different countries such as Portugal and Finland, European and OECD members with similar HE systems, provides a twofold advantage. First, they can actually work as a case study because, despite their contrasts, they are relatively small and peripheral European countries, and their HE systems are both binary (Kauko and Diogo 2012; Diogo 2016) (constituted by polytechnics and universities in Portugal and universities of applied sciences and universities in Finland). Second, despite the political and cultural differences, common external mechanisms have been used in both countries, affecting the adoption of reform ideas (Peters 1997). Thus, by taking the specific case of the HE sector, this paper intends to reflect on the role that intergovernmental organisations, namely the OECD, have in influencing or presenting *neoliberal solutions* in HE systems, using as examples the 2007 and 2009 reports for Portugal and Finland, respectively, and their translation in national policies.

## 2. Neoliberalism in Higher Education Policies

Although neoliberalism's specific nature is contested (Krugman 2009; Stiglitz 2007; Ong 2006; Shamir 2008), there is a common view of it as an economic doctrine and an economic discourse or philosophy which has become dominant and effective in world economic relations as a consequence of superpower sponsorship (Olssen and Peters 2005). Within the economic sphere, neoliberalism is linked to globalization, especially concerning freedom of commerce and trade. However, neoliberalism represents only one dimension of globalisation (Olssen and Peters 2005), and it is not limited to an economic doctrine but also to "(...) a complex, often incoherent, unstable and even contradictory set of practices that are organized around a certain imagination of the 'market' as a basis for 'the universalization of market-based social relations" (Shamir 2008, p. 3). In such an epistemological perspective, neoliberalism is interpreted as a biopolitical mode of governing (Ong 2006), meaning it is not only about the economy or globalisation, but also about human relations. In fact, Ong (2006) looks at neoliberalism as a flexible "technology of governing" used *enthusiastically* and differently by different countries and regimes according to their political administrative traditions (be they authoritarian, democratic or communist) in order to compete in the global economy.

Within this theoretical framework, the dominant notions of framing public reforms in current times as governance (Ong 2006) or a knowledge economy are embedded within neoliberal ideas, which are sustained in three main *policy technologies*: market, management (or managerialism) and performance (or performativity) (Ball 2003, 2016). Market technologies are mainly associated with several arrangements of competition and rational choice and exogenous and endogenous modes of privatisation (which may happen simultaneously) which create market or quasi-market realities, promoting the increasing withdraw of the state from providing public services. Endogenous privatization introduces the market into the public sector through choice and competition, creating a direct

relationship between consumer preferences and institutional wellbeing with the purpose of making public service organisations more business-like (Ball 2016, p. 1049). In turn, exogenous privatization brings new providers into the educational service delivery market (e.g., consultancy services). In England, for example, Ball (2016, p. 1049) remarks that the present debate is not about the identity of the state schooling providers and who they are but rather if these providers should be able to profit directly from such provisions. These privatizations, together with the other policy components (management and performance), symbolize much of the neoliberal "modernization" of the state, and what other scholars have called the "hollowing out of the state" (Bovens et al. 2002; Pollitt and Bouckaert 2011; Hooghe and Marks 2001), characterised by the increasing use of contracting out (Ball 2016).

Management (or managerialism) is associated with new power relations, social connections and less democratic and less caring attitudes, as well as what Ball (2016, p. 1049) calls "methods for reculturing educational organisations". These technologies of reform do not impose behaviour, but they coerce professionals to do things differently, creating new roles, opportunities, values (Chatelain-Ponroy et al. 2018), discourses, vocabularies and ideas that, when not enthusiastically accepted, label professionals as unprofessional, irrational or even archaic (Ball 2016, p. 1049). This is intimately linked with the concept of performativity (performance management), which relates to accountability agendas, new and complex systems and indicators of performativity associated with the *new order* of doing things through measurement and comparison techniques (Ball 2016, p. 1050).

Managerialist ideologies and New Public Management (NPM) have been instruments used to put in place these three technologies. Despite differences in models and enforcement levels, the core idea, associated with NPM and managerialism, is to do more with less under the belief that the government will function better when it adopts private sector management practices instead of the old bureaucratic model of organisation. Competition—both among organisations and individuals–thus works as the fuel to make public organisations reduce their costs and perform their jobs better.

In the last few decades, NPM and managerialism, as a *model* for more efficient, effective and less expensive governments, has been identified as one of the most *effective* convergent tendencies architecting reforms in distinct sectors of public administration (Van de Walle and Hammerschmid 2011). These (forms of) convergent trends, very much sponsored by the phenomena of globalisation, internationalisation and Europeanisation—as transnational processes—are visibly empowered and influenced by international organisations such as the EU (Featherstone and George 2014) and the OECD, or what has been recalled as "forceful agents of policy transfer" (Stone et al. 2019, p. 7). Nevertheless, this does not imply that policy transfer and borrowing do not assume a straightforward convergence. On this, one can also draw on the work of institutionalist scholars such as DiMaggio and Powell (1983), who identified three specific mechanisms leading to convergence: coercive, mimetic and normative isomorphism. "Coercive factors involved political pressures and the force of the state, providing regulatory oversight and control; normative factors stemmed from the potent influence of the professions and the role of education; and mimetic forces drew on habitual, taken-for-granted responses to circumstances of uncertainty" (Powell and Colyvas 2007, p. 2). Identities and interests change as a result of particular localisms (de Sousa Santos 2006) or "localised globalisms" (de Sousa Santos 2006), which might be consubstantiated in shifts in social norms, values and beliefs (Featherstone and George 2014), as well as shifts in cultural (e.g., new ideas about knowledge) and economic factors (e.g., decline of public funding) (Deem 2001; Diogo 2014), and these may occur in response to transnational or global pressures only loosely connected to international organisations. Thus, although policy transfer and convergence can circulate internationally, there is always the mediating influence of the local level to shape, contest, negotiate, etc. how these local manifestations occur (Ong 2006). Moreover, or reinforcing this idea, looking at domestic adaptation through the lens or perspective of the power of domestic actors allows us to identify the potential significance of divergences between different national

settings (Featherstone and George 2014), enhancing comparisons between such countries as Portugal and Finland.

Convergence is assumed here to be the process of having similarity of practices, objectives and instruments, resulting in the tendency of societies to become more alike by developing similar structures, processes and performances (Heinze and Knill 2008). Policy convergence is linked to and enabled by policy diffusion as the process that occurs "when government policy decisions in a given country are systematically conditioned by prior policy choices made in other countries" (Simmons et al. 2006, p. 787; Vögtle 2014). Policy diffusion is then framed by globalisation, internationalisation and Europeanisation, usually assumed as movements that promote convergence at the economic, political, social and cultural levels, being part of the neoliberal rhetoric and practice, as tools to reform the public sector are, to a great extent, a product of policy diffusion and policy transfer. Although each country has been developing some specificities in their policy developments, with many due to cultural and historical traditions, "it is extremely improbable to think that all these disparate political systems would have concocted the same ideas about change at the same time" (Peters 1997, p. 71).

Bearing this in mind, it is relevant to acknowledge the role of intergovernmental organisations in promoting isomorphic trends when spreading neoliberal ideas and influencing the definition of new legal frameworks for the public sector (Dimitrakopoulos and Passas 2003; Laegreid and Christensen 2013). Among these intergovernmental organisations, the OECD seems to play a relevant role in framing HE policies around the world (Kauko and Diogo 2012; Rinne et al. 2004; Kallo and Semchenko 2016; Kallo 2009; Saarinen 2008). Today, the OECD has acquired a "brand identity" among its signatory countries by constructing and disseminating knowledge about various HE issues (e.g., documenting and forecasting trends, providing educational indicators and supporting forums where stakeholders come together) (Shahjahan 2012, 2013). According to Shahjahan (2013), the OECD develops an imperial logic by using soft power. By constructing and disseminating knowledge about HE, the OECD frames global educational policies in a colonial logic. However, despite the OECD's publications on several HE systems, there is scant research on its influence on national HE discourses, legislation and policy implementation (Shahjahan 2013, p. 677). Some exceptions are found in the studies of Kallo (2009), Kallo and Semchenko (2016), Shahjahan and Torres (2011), Martens et al. (2007), Kauko and Diogo (2012), Saarinen (2008) and more recently, the study of Pettersson et al. (2017) featuring Norway and Sweden. Analysing the OECD's influence in spreading neoliberal reforms in HE is particularly relevant since this institution has no coercive power, as it is not able to impose any new legislation but only to produce recommendations.

It is possible to argue that neoliberalism is internationally enforced in the public sector through mechanisms of *soft law* as, for example, country reviews and recommendations from intergovernmental organisations that are not limited to the OECD but include others such as the EU, the World Bank and the United Nations (UN) (Stone et al. 2019). These intergovernmental institutions work in networks as epistemic communities and coordinators in spreading reforms and are expected to widely diffuse administrative innovations (Vögtle 2014; Dolowitz and Marsh 1996, 2000; Evans 2009; Haas 1992).

The historical, cultural and national policy specifics and structural characteristics of politico-administrative systems (Bleiklie and Michelsen 2013) are considered powerful factors in explaining differences in policy design and implementation processes, as well as national outcomes. This corroborates the ideas from Czarniawska and Sevón (1996) on the processes of law implementation and enforcement being much more related to cultural translation or localised adaptation rather than adaptation of reform. Additionally, Ball (1998, 2012) argues that policy design differs from policy formulation and implementation, at least at the national level, where "policy-making is a process of *bricolage*: a matter of borrowing and copying bits and bits of ideas from elsewhere" (Ball 1998, p. 126). In turn, this process of looking at the *neighbour* and applying those solutions, policies and practices

that seem to be more successful not only contributes to increased convergence but also isomorphic patterns of action.

Bearing such perspectives in mind, as well as the concerns of Bleiklie and Michelsen (2013, p. 113) about the gap in "questions about policy making, such as how and by whom HE policies are designed", this paper contributes to bridging this gap by analysing how the OECD, by being positioned as a key actor in national HE policy making, enforces the neoliberal credo both in Northern and Southern Europe, taking Finland and Portugal as examples.

## 3. Overview of the Portuguese and Finnish Higher Education Systems

Higher education institutions (HEIs) are characterised by specific cultural features inherited from the past, shaping their paths (Simola et al. 2013) and the ways they respond to current challenges (Vaira 2004, p. 494). Examples of these responses or localisms are the paths that both HE systems developed and sustained in their different welfare states and the structural characteristics of their political-administrative systems. Both countries' path dependences have been substantially different, with political and economic developments *shaping* the "basic contours of social life" (Simola et al. 2013, p. 614). For example, while Portugal received external economic support three times in the last four decades (the last time was in 2011), Finland became one of the most robust European economies and built a typical Nordic welfare state (Esping-Andersen 1990; Välimaa 2004), emphasizing HE as a public good (Ylijoki 2014) and equality as its core value (Välimaa 2019).

Both in Portugal and in Finland, HE was, for many years, an elite system. However, at an earlier stage than Portugal, Finnish HE expanded rapidly during the 1960s, a process related to and a result of a welfare state agenda supported by the major political parties (Välimaa 2001). In Portugal, it was only after the 1974 revolution that HE evolved into a mass HE system (Amaral and Carvalho 2003). As in other European countries also framed by the Humboldtian or Napoleonic models, both in Finland and in Portugal, HE has been considered crucially important in the creation of a national identity and in building the nation, as the social elite and its associated professional groups are trained in public HEIs and employed in the service of the state (Välimaa 2001; Rinne 2004).

From the 1960s to present day, Finland has been able to create, develop and maintain the idea of equality as one of the structuring principles of Finnish society (i.e., equal educational opportunities for all citizens regardless of their gender, socioeconomic status or location) as well as a very positive attitude towards education throughout Finnish history (Välimaa 2001, 2004). Universities, university degrees, teachers and professors still retain a high social prestige in Finland (Välimaa 2001). Quite different is the situation in Portugal, where the value of a university degree and the teaching profession has depreciated (Almeida and Vieira 2012).

The New University Act (2009) consolidated the path of educational equality, as the legislation stipulated that education leading to a degree in Finnish universities is free of charge (*Yliopistolaki* 558/2009 §1, 8). Gradually, Finnish universities were given increasing autonomy through Law 645/1997 (26 July), and in 2006, also as part of NPM efforts, a structural development programme was introduced aimed at dropping the number of Finnish HEIs within the next 10–15 years (OKM 2016). At present, Finland has 13 universities and 22 polytechnics, which have recently been renamed as universities of applied sciences (UAS) (OKM 2012). In turn, the Portuguese network of HEIs comprises 38 universities (14 public and 24 private universities) and 65 polytechnics (20 public and 45 private polytechnics) (Pedrosa 2017). The number of enrolments in Portuguese HEIs at both universities and polytechnics has been quite steady, showing an overall increase since the beginning of the new millennium, even during the period of the financial crisis that started in 2008. Public HEIs tend to be the first choice of the majority of Portuguese students (Oliveira and Soares 2016).

In both countries, intergovernmental organisations, and more specifically the OECD, have been assuming an important role in HE policy design and subsequent implementation.

In Portugal, political actions from the government and academics, combined with the support and guidance of intergovernmental organisations (especially the World Bank, the OECD and the EU later), contributed to the development of the national HE system through the creation of the vocational and private subsectors through the 1970s (Amaral et al. 2003). Portugal was one of the former OEEC founding members in 1948 and ratified the OECD convention on the 4 August 1961. Since then, and especially during the 1970s, through the action of the Minister Veiga Simão, who was very much inspired by human capital theories and the OECD commandments, Portuguese HE was democratised (Amaral et al. 2002), fulfilling a function of social improvement (Stoer 1983, p. 818). Additionally, in Finland, the creation of a binary system in the mid-1990s through the establishment of polytechnics was catalysed by the OECD (Kauko and Diogo 2012), although the process differed substantially, as political, economic and cultural circumstances were also different.

Finland joined the OECD on 28 January 1969, and in 1992, the OECD made the first evaluation of Finnish educational policy (Rinne 2004). Several scholars have been consensual about the fact that not only has Finland been the EU's star pupil concerning the implementation of HE policies, but it has also been exemplarily devoted to the OECD's recommendations (Kivinen and Kaipainen 2012).

Referring to the influence of the OECD in Portuguese policies, Sacuntala de Miranda (1981 in Teodoro 2000, p. 54) used the expression *ocedeísmo* to describe the diffusion of an educational ideology based on the active participation of the country in OECD projects. In the views of Stoer (1983) and Amaral et al. (2002), it is valid to say that the need to catch up with European standards, together with the OECD and the World Bank's influence (and even the IMF), caused governmental policies to reflect a steady increase in the functionalisation of education in general and HE in particular with respect to issues related with the country's economic development (Amaral et al. 2002, p. 7). This *functionalism trend* in HE and its purpose have been reinforced by pressures to change the way knowledge, training and education are provided, which mainly emerged with the popularity of the "knowledge society/economy" expression, combined with other *managerialist* trends, namely neoliberal policies in the mid-1990s and the increasing influence of neoliberalism over the past decade (Lima 2012). Additionally, as member countries had to fulfil the economic and political criteria of the EU, the launching of a single European currency gave member states a new philosophy, strengthening a neoliberal economic trend (Sousa and Fino 2007, p. 608).

In 2006, at the request of the government, the OECD and the European Network of Quality Assurance in HE (ENQA) conducted an extensive review of the Portuguese HE system with the purpose of reforming the system while adopting the European guidelines approved in the context of the Bologna process (OECD 2007). In a similar way, the OECD (2009) also published a country review of Finnish HE, resulting in the New Universities Act (Law 558/2009), and later in 2014, a similar legislative update for the non-university sector was drafted: Law 932/2014 defines the Universities of Applied Sciences Act for Finland by *polytechnics (UAS now).*

The Nordic practices of publicly funded schooling and inclusive, comprehensive school systems still persist, which are quite different from Southern Europe, but globalisation and Europeanisation have influenced Finnish politics and administration by installing pressures to increase competitiveness in the public sector. In the words of Pettersson et al. (2017, p. 723), Scandinavian values are challenged by international ideals of competition and outcomes orientation (i.e., successful globalised localisms).

## 4. Materials and Methods

To understand if and how the OECD promotes the trilogy of neoliberal components in different countries while simultaneously creating convergence in HE policies, this paper relied on intensive document analysis. More specifically, the authors cross-compared both OECD countries' reports on the 2007 Portuguese and 2009 Finnish HE systems and performed the same comparative analysis with the (new) legal frameworks for HE in both countries, which were issued after these OECD evaluations were performed.

Based on the literature review and the theoretical contributions, both the OECD reports were submitted to content analysis through thematic coding with the help of Nvivo11.4.1 qualitative data analysis software. Three main categories of analysis were considered to drive our analysis through the documents: performativity, market and management (Ball 2016). To assess the extent to which the OECD enforces a neoliberal ideology in both Portuguese and Finnish HE systems, the same procedures were applied to the national legislation in both countries, searching for distinctive elements in the reports that corresponded to these three main categories. Table A1 summarises these elements in both OECD reports for the Portuguese and Finnish HE systems, and Table A2 summarises how these elements have been translated in the national legislation in both countries (Appendix A).

## 5. Data Discussion and Comparative Analysis: The Neoliberalism Triangle

### 5.1. The Market

The market *corner* is associated with several arrangements of competition and rational choice as well as exogenous and endogenous modes of privatization, creating market or quasi-market realities. Ultimately, such behaviour tends to gradually contribute to withdraw of the role of the state in providing public services. Examples of the presence of market elements in the OECD reports of both countries relate to the legal status of HEIs, to the commercialisation of HEI research and development (R&D) and tuition fees.

The OECD review teams have identified distinct problems in both the Portuguese and Finnish higher systems, but both panels agreed that at the national and institutional levels, there was a need for a more professional approach to management driven by goals and outcomes (OECD 2007, pp. 132–33). In this way, the main *instructions* from the Portuguese OECD report (2007) were that HEIs, although still supported financially by the government, should operate according to the private sector rules and generate more and diversified income sources (OECD 2007, p. 66) through exogenous and endogenous modes of privatization. For example, the OECD prescribed a change to HEIs' legal status, which could vary between a university foundation or a "new type of public entity" (p. 67), which in Portugal assumed the form of a public institute (cf. Table A1):

"The new legislation should establish institutions as self-governing foundations. Still supported financially by government, they would operate within the private sector" (OECD 2007, p. 141).

In the case of Finland, the OECD review team pointed to the lack of autonomy and entrepreneurial character of Finnish HEIs (OECD 2009, p. 58), and similar to the Portuguese case, the OECD (2009, p. 108) recommended Finnish universities to become non-profit corporations or foundations under private law:

"It seems very appropriate to redefine the HEI (both polytechnics and universities) as so-called "Legal persons", rather than as civil servant units.

Within this approach, there are alternatives for institutions:

- As non-profit corporations
- As foundations" (OECD 2009, p. 108).

Despite clear historic, geographic, cultural and economic contrasts, both OECD reports advised a change of HEIs' legal status, suggesting the establishment of institutions as *self-governing foundations* still financially supported by the state while simultaneously urging them to generate more income and diversify their funding bases. This recommendation is clearly framed with neoliberalism and even managerialism. Using Stephan Ball (2016) terms, one can classify the recommendation for the creation of foundations in both countries as endogenous privatization because the aim was "( . . . ) to make public service organizations more business-like and more like businesses" (Ball 2016, p. 1049). In fact, despite their increased private character, HEIs in both countries still belong to the public sphere domain, as the state remains the main funding provider. However, the OECD is keen in legitimising its position concerning the shift in HEIs' legal status under the constant argument of enhanced and clearly defined institutional autonomy. "Government should introduce comprehensive university and polytechnic legislation in which the autonomy of institutions is clearly defined." (OECD 2007, p. 67).

In line with the OECD's recommendations, the Portuguese government approved the RJIES (Law 62/2007), which became the new legal regime for HEIs. This law thus aligns the statutes relative to public and private institutions, universities and polytechnics. It updates legislation concerning public and private universities and polytechnics' autonomy and the legal regime governing HE quality and development as well.

The same holds true for the Finnish HE system. The New Universities Act established universities as *independent legal personalities*, either as public universities or foundation universities, although the state, such as it happens in Portugal, still remains the main source of funding. The relationship between HEIs and the government is thus the core of the new legislation, which has been redefined by following closely the 2009 OECD report (Kauko and Diogo 2012).

In fact, for both Portuguese and Finnish HE systems, the possibility given to HEIs to choose to remain a public institute or to become a public foundation under private law is something totally new, allowing the cultural translation (and incorporation) of more market and management mechanisms in governing the public sector in both countries. Nevertheless, it seems that the new narratives were not automatically or fully incorporated into actors' subjectivity. Thus far, only five universities in Portugal have chosen this path, and only two Finnish universities adopted the foundational model, which is not *representative* of the Finnish reality. Slightly different from the Portuguese scenario, the New Universities Act is seen more as a continuity in Finnish HE policy than a reform, and emphasis was not so much placed on the shift in universities' legal status (as only two universities adopted this model) but rather on the *discharge* of the civil service designation from all employees of HEIs (approached in the next section). Also dissimilar to Portugal, the path of HEI institutional autonomy is older when compared with their Finnish counterparts (Diogo 2016). In this sense, it can be argued that Finnish decision makers correlate the introduction of more market elements in their HE system with the process of increasing institutional autonomy, even if the OECD evidences concerns as to whether HEIs will be able to manage autonomy effectively (OECD 2009, p. 107). Aligned with such concerns, the (perceived) possibility of increasing institutional autonomy regarding the management of financial and human resources that university foundations (seemed to) enjoy, allowing for greater financial predictability and stability, is also used in the OECD (2007) narratives. "We believe that government must disengage from the detailed control of the system and must give the institutions greater freedom to regulate themselves and to innovate" (2007, p. 140). In line with the literature (Martens et al. 2007; Simola et al. 2013; Kauko and Diogo 2012), it seems thus possible to claim that that the OECD has the power to determine or strongly influence professionals' and policy makers' ideas, as well as the drafting of national legislation, as the 2007 and 2009 OECD recommendations and assumptions on the new legal status for HEIs for both countries evidence (OECD 2007, p. 67; Tables A1 and A2 in Appendix A).

Another example of market initiatives (endogenous privatization) instigated by the OECD relates to the recommendation for Portugal to increase tuition fees "in order to help provide additional resources to the institutions and to acknowledge the significant positive financial advantages that a HE qualification confers on graduates throughout their working life" (2007, p. 12). In 2007, tuition fees were significantly increased in Portugal, whereas in Finland, the OECD suggested several times that HEIs should charge tuition fees (2009, p. 68) at least to international students to "provide additional resources and incentives for institutions to internationalise", among other reasons. The OECD (2009) also suggested more funding diversification so that a "more entrepreneurial ethos" could emerge, as well as an "evolution of the planning agreement model, as a prime instrument of accountability" (OECD 2009, p. 115). A diversification of the funding basis would sustain "the relationship between a planned system (of which Finland is a very good example) and a market economy for HE ( . . . )" (OECD 2009, p. 115). Another example of these market initiatives in HE *suggested* by the OECD targets the commercialisation of R&D initiatives. "The enhancement of the impact and commercialisation of HEI R&D and educational and research services by the setting up of joint enterprises" (OECD 2009, p. 14).

In this renewed *planned market and management system*, both the RJIES and the New Universities Act promote a loose legislative framework, leaving much room for each university regarding decision making and management practices so that they can respond "more flexibly and independently to the challenges arising from their new financial status" (OKM 2016).

## 5.2. Performativity

The element of performativity or performance management is profoundly related to accountability agendas, new and complex systems and indicators of performativity, usually associated with the *new order* of doing things and mostly based on measurement and comparison techniques (Ball 2016, p. 1050).

Performativity and management elements are strongly incentivised by the OECD discourse, mostly being visible in accountability, assessment and performance management techniques and indicators, as well as shifts in governing bodies' composition and *modus operandi*. A good example is the following quote: "The shift in steering philosophy would be supported by the system of institutional performance agreements or contracts negotiated between the ministry and individual institutions" (OECD 2007, p. 141). This sentence translates one of the main features of an effect of performativity identified by Ball (2016, p. 1055) as the trend to replace commitment with contracts. Nevertheless, this new system of institutional control based in contractual relations with the state did not translate into the highly desired institutional autonomy, at least in Portugal. Considering the economic crisis in which the country has been immersed in the last decade, the austerity measures imposed by the bailout and the successive changes determined by the treasury that have already affected many of the benefits that led these institutions to opt for the foundational model, this so-called (financial) autonomy that appealed so much to HEIs might have been useless. Simply put, the advantageous framework that apparently would exist in managing according to private law seemed to get lost in a scenario of a weak institutional and economic path. For the Finnish HE sector, however, the OECD tends to be more *prescriptive*, as Finland has a more recent tradition in institutional autonomy than Portugal. "In the event of the adoption of the "legal person" principle, it would be necessary to establish a governing body or board of trustees for each type of HEI accountable to government" (OECD 2009, p. 108). In addition, those boards or governing bodies should behave as follows: "The most effective boards are those that have developed a board code of conduct and that practice regular self-assessment of their own performance as a board" (OECD 2009, p. 155). It seems that the OECD proposed a selection of the "best" institutions to assure that the new model would be successful. However, the extent to which these proposed changes were incorporated by HEIs might be subject to paradox interpretations. If, on one hand, financial and management autonomy results in a step

forward in terms of independency from the state, the state remains the main funding lender of HEIs. The OECD was thus quite straightforward in its recommendations for the Finnish HE regarding this financial autonomy. "By definition, the "legal person" would assume significant devolved responsibility from government over a range of domains e.g., investment, property, share-buying, etc." (OECD 2009, p. 108).

Shifts in institutional governance and management have been a general trend of many continental European nations, with the state introducing regulatory mechanisms similar to market ones, which are used as public policy instruments (Amaral et al. 2002; Ball 2016). *Steering from a distance* is justified by the perception that it is impossible for the state to efficiently regulate the processes at the system and institutional levels within a "centralised control" logic (Amaral et al. 2003, p. 137). Thus, through this, the state not only aims at creating inter-institutional competitiveness mechanisms that increase institutional efficiency but also mechanisms that make institutions more responsive to external calls, namely when these are bonded to economy needs. Competition, efficacy and efficiency are all part of the performative regime (Ball 2003).

With respect to accountability elements and management indicators, the OECD legitimises changes in the traditional models of university governance, with the increasing need for accountability requirements and their contribution to improve HEIs' efficiency and strategic decision making (OECD 2007, pp. 128–29, 157; OECD 2009, p. 78; cf. Tables A1 and A2 in Appendix A).

"If we are serious about driving innovation in our universities (and therefore in our nations), governments should disengage themselves from the details of university operations, yet maintain clear accountability requirements. Pressure to change the traditional models of university governance has become more acute in recent years as public funding has often become more targeted ( . . . ), institutional autonomy has increased, and external performance management and other accountability mechanisms have required universities to publicly demonstrate their efficiency and effectiveness" (OECD 2007, p. 157).

The OECD has also been keen on inciting competition in both HE systems by means of performance agreements, recommending that "management has to exercise judgement and a firm hand in subsequent internal distribution in a micropolitical competitive faculty environment" (OECD 2009, pp. 82–83).

Being quite general in character, the analysis of the OECD reports confirms the idea that the organisation creates a setting where performance and managerialism elements, such as "( . . . ) evaluations and assessments appear natural, self-evident and rational as well as highly adaptable to national settings" (Pettersson et al. 2017, p. 735), even when these settings have walked dissimilar paths and localisms.

*5.3. Management*

Management or managerialism is associated with "methods for reculturing educational organisations" (Ball 2016, p. 1049), such as new power relations and social connections and less democratic and less caring attitudes. These neoliberal technologies tend to coerce professionals to do things differently, creating new roles, values, discourses, vocabularies and ideas that, when not enthusiastically accepted and incorporated, label professionals as unprofessional, irrational or even archaic (Ball 2016, p. 1049). In the OECD reports, such elements are visible in changes related to human resource practices, institutional governance and management *ethos* and practices, taking into account the need to increase efficiency and quality and the pursuit of excellence (cf. Appendix A).

In terms of staff management, for example, the OECD also set management and performativity policies by recommending for both HE systems the removal of the civil service designation from all employees of HEIs (OECD 2007, p. 141; OECD 2009, p. 90), a suggestion that both countries incorporated in the legislation drafted after the OECD reports (cf. Law 558/2009 and Law 62/2007) (Table A2). However, more components of the neoliberal triangle are sponsored in terms of institutional governance and management, namely through changes in the composition and nature of institutional governing bodies.

Briefly, both OECD reports recommended fewer—or even extinguishing—collegial bodies and for their members to streamline decision-making processes to include external stakeholders in governing bodies and strengthen leadership. To this end, although the OECD's recommendations were *coordinated*, the OECD was harsher on Portugal than Finland:

> "The slowness of decision making, and the lack of clarity and transparency of decisions when they are made, seem to be an inevitable consequence of the structuresand are totally out of step with modern and more effective approaches. Collegiality is a valuable concept in a HE institution but we believe that this can be achieved within a framework of a reduced number of layers of decision taking, and with bodies with much smaller membership" (OECD 2007, p. 66).

> "The evolution from a Humboldtian model to one of a modern university, with multiple objectives, diversified funding, purposive steering mechanisms and a strong external responsiveness is thus, in our opinion inevitable" (OECD 2009, p. 114).

Recommendations like these culminated in a shift from a collegial governance model to a more managerial one where it is possible to expect *less* resistance in Finland than in Portugal. This is explained by the fact that the government is still the main funder of HE, along with the substantial increase in institutional autonomy HEIs have been gaining. Additionally, HE and the academic or teaching profession still hold high-value positions in Finland, strengthened by the strong role unions hold. Finnish universities now have procedural autonomy in deciding how to reach the targets set by the ministry to hire their own personnel and to determine the powers of administrative bodies (Law 558/2009, Table A2). Their size and membership are, however, determined by law, and the new governing and management structures created new positions in the university, such as financial officers and a joint committee on financial matters including outside experts from business (Diogo 2016), which is typical from the business world and represents a shift in the way Finnish HE is now steered, namely focusing more on professional management with performativity elements and being closer to the market. Performativity might thus enhance micromanagerialism or the expectation of an increase in the "micro-politics of little fears", as Lazzarato put it (Lazzarato 2009, p. 120).

In turn, and quite interesting, are the OECD recommendations for Portugal (cf. Appendix A). While being "stuck" by the Portuguese micromanagement of the system (e.g., "detailed control over new programmes and modifications to existing programmes" (OECD 2007, p. 64), the OECD recommends a "(...) disengagement of the government in Portugal from the detail of institutional management and control to one of overview and strategic vision" (OECD 2007, p. 64).

One should remember that NPM opposes collegial governance modes once that collegial governance and professional power are considered inadequate to the needs of competition induced by economic globalisation and by the knowledge society (Carvalho 2012).

It is also possible to infer that the emergence of "new actors" in the governance structure of HEIs brought new values and norms to their cultural-cognitive framework. This is completely aligned with the performativity and management components that HEIs and their professionals have to embody to be rated as excellent and successful (Ball 2003, 2016).

When analysing the influence of OECD in the developments and production of national HE legislation, while bearing in mind the cultural and historic contexts of both countries, it is acknowledgeable that the organisation's reports act as a legitimisation source for reforms, but it seems fair to believe that the OECD's influence in Portugal in this domain has been more substantial and direct than in Finland. This can be explained by the fact that Finland is a younger country than Portugal and joined the EU later (in 1995), and therefore, the practice of international influence in HE policies is not as old there as it is in Portugal. Another possible explanation relates to the distinct economic situations of both countries, as proven by their positions in the UNDP.

### 6. The *Versatile* Normative Side of the OECD: Convergence in Translating Neoliberal Recommendations

The content analysis of both OECD reports demonstrates that even if dissimilar problems are diagnosed, similar recommendations are proposed which are aligned with neoliberalism ideas, revealing the ideological nature of the recommended policies. Although Portuguese and Finnish reports were produced by different "academics", they come to suggest common responses (Torres and Schugurensky 2002) which are, in fact, not only common but also different problems. With this, it is clear that the OECD has been playing a relevant role as a transfer agent in the internationalisation of HE policies. This is not new, since previous studies already identified the OECD trend to influence national policies, especially in Portugal (Amaral and Neave 2009; Kauko and Diogo 2012; Saarinen 2008; Kallo 2021). By incorporating, translating, legitimating and disseminating the wider *rationalised myths*, related challenges, ends and means as an objective reality (Vaira 2004), the OECD develops a general and common framework defining the new context and imperatives in which HEIs have to operate nowadays. By acting as dissemination agencies on a global scale, "(...) they contribute to construct and structure a de-localized and global organizational field (Powell and DiMaggio 1991) where national higher education policies and institutions have to face and operate" (Vaira 2004, p. 488).

As both the Finnish and Portuguese cases exemplify, both reports do not differ much in their recommendations for the trilogy of neoliberal components proposed by Ball (2016). By evidencing the NPM rhetoric and managerialist ethos in the solutions presented, these findings not only corroborate Ball's (2016) views on the *slouching beast* but also the claims of Amaral and Neave (2009) that "the OECD contributes directly to disseminating neoliberalism not only by showing that the doctrine works, but that it is an appropriate framework within which plausible solutions may be sought, identified and acted upon". Furthermore, as the proposed solutions are quite similar for both countries, and since the OECD cannot exert coercive power, one can say that the professional recommendations can be assumed as normative isomorphic pressures. This, in turn, provides national governments with increased manoeuvres to legitimate reforms and to install new regulatory frameworks for their HE systems, with some political malleability resulting from national specifics. Examples include Law 62/2007, set for the Portuguese HE system after the publication of the OECD report, and Law 558/2009, further extending the autonomy of universities. However, it also true that this study focused on these countries, and the results might have been different if this exercise were performed with other national realities (e.g., not small or peripherical countries, countries having unitary HE systems and non-European countries).

Although the OECD recommendations cannot be assumed as the only factor imposing neoliberal principles in different countries, it is striking how two different European countries translated them into very similar or convergent domestic legal solutions. Empirical analysis based on content analysis of OCDE recommendations and national legislation allows for concluding that through its peer reviews, recommendations and indicator studies, the OECD already plays a *soft law* role in these HE systems, one that is in fact similar to other international organisations such as the European Commission, and its impositions and recommendations promote convergence in policy design and policy making at the national level.

The normative framework within which the OECD operates, mainly being economic driven in its nature, promotes an orientation to the market, leading to gradual disaggregation of the public HE system through means of (endogenous and exogenous) privatisation, management (efficiency, quality and excellence) and performativity elements (Ball 2016). Using Shahjahan's (2013) terms, it is possible to say that the OECD reports colonised both Portuguese and Finnish HE policies with neoliberal (imperial) narratives and practices. Even without coercive capacity, normative isomorphism played by intergovernmental organisations and their *creeping* influence works effectively in spreading neoliberalism, although with different degrees of enthusiasm in Portugal and Finland, as the national leg-

islation evidenced (Table A2). Notwithstanding this, while international convergent forces, similar instruments and mimicry drive countries to adopt similar HE practices, national characteristics allow for different policy implementation processes and outcomes. Such a finding triggers an interesting discussion when cross-comparing cultural data and policies: how much of the similarities that have been occurring in both Portuguese and Finnish HE systems are outcomes of policy diffusion and policy transfer? This paper argues that both OECD reports and subsequent pieces of legislation that originated from them—the Portuguese RJIES and the Finnish New University Act—can represent a successful example of policy travel and translation of neoliberalism to and by HE systems and institutions. This study also confirms that there is no single model for effective public management reforms, but the "one size fits all" recipe is diffused to countries apparently as different as Portugal and Finland (Diogo et al. 2019; Takala et al. 2018). Data analysis brings a relevant contribution to understanding how HE policies in the north and south European context are becoming more similar than different under the influence of international organisations. In the words of Rinne (2004, p. 127), Finnish HE policy "is no longer very national nor very Nordic, but more and more EU and OECD-like", a common fact to small nations like Finland (and Portugal) as they are not fully able of carrying out their own independent foreign, domestic or even educational policies. Further research implies dissecting the latest OECD reports for both of these HE systems to track the continuity of these trends and to enlarge the pool of countries to compare. Another line of future endeavour lays in cross-sector comparison to analyse the OECD's influence in other sectors in addition to HE, such as the health sector.

**Author Contributions:** T.C. and S.M.D. had the idea for the paper, and both authors conducted the empirical and comparative analysis of the OECD reports and national legislation. All authors have read and agreed to the published version of the manuscript.

**Funding:** This work was supported by Fundação para a Ciência e a Tecnologia (grant number POCI-01-0145-FEDER-029427).

**Data Availability Statement:** Data supporting the reported results can be found in both OECD reports referred to in this paper.

**Acknowledgments:** The authors are thankful to the reviewers of this manuscript. Part of this study was developed under S.M.D doctoral thesis., supervised by Alberto Amaral and co-supervised by T.C. and Jussi Välimaa. The authors are thankful to Jussi Välimaa, former PhD supervisor of Sara Diogo and to the reviewers of this manuscript.

**Conflicts of Interest:** The authors declare no conflict of interest. The funders had no role in the design of the study; in the collection, analyses, or interpretation of data; in the writing of the manuscript, or in the decision to publish the results.

## Appendix A

**Table A1.** Examples on how the OECD proposes neoliberal reforms for the Portuguese and Finnish higher education (HE) systems based on three main components of neoliberalism: market, management and performativity (Ball 2016).

| Neoliberal Triangle | Finland (2009) | Portugal (2007) |
|---|---|---|
| entry 1 | data | data |
| entry 2 | data | data |
| Market | **Tuition Fees:** "In terms of resource management, refinement of a funding formula based more on the attainment of results and outputs, and the consideration of tuition fees for international students" (p. 15). + "Whilst other systems have espoused tuition fees with varying degrees of enthusiasm and reluctance, our widespread discussions with parliamentarians, stakeholders, students and institutions suggested that few Finns believe that a larger private financing initiative through student tuition fees should be introduced into the system" (p. 86). + "Higher education institutions should be permitted to charge fees" (p. 87). **Legal Status:** "The dual system which the review team encountered is thus held to be clearly differentiated in terms of: ( … ) the model of governance and administration—state compared with a municipality, joint municipal bodies, or foundations ( … )" (p. 13). + "It seems very appropriate to redefine the HEI (both polytechnics and universities) as so-called "Legal persons", rather than as civil servant units. Within this approach, there are alternatives for institutions: − As non-profit corporations − As foundations" (108). | **Tuition Fees:** "Tuition charges should be increased significantly, in order to help provide additional resources to the institutions and to acknowledge the significant positive financial advantages that a higher education qualification confers on graduates throughout their working life" (p. 12). "Tuition charges for postgraduate students should be deregulated and allowed to increase to rates closer to the full costs of the programmes" (p. 144). **Legal Status:** "Government should introduce comprehensive university and polytechnic legislation in which the autonomy of institutions is clearly defined." (p. 67). + "These considerations (underlined by many remarks to the review team) call for new legislation governing the higher education institutions. The new legislation should establish institutions as self-governing foundations. Still supported financially by government, they would operate within the private sector. They would have managerial freedom and finances separately accounted for outside the state system. The civil service designation would be removed from all employees of the higher education institutions. The institutions must satisfy government that they are prepared to accept this freedom and that they are willing to confront the difficult leadership and managerial decisions that are an inherent part of such an arrangement" (p. 140). |

**Table A1.** *Cont.*

| Neoliberal Triangle | Finland (2009) | Portugal (2007) |
| --- | --- | --- |
| **Management** | "The internal cultures, organisational and management practices which on the one hand reflects a collegial and Humboldtian model (universities), and on the other, a more managerial/corporatist model (polytechnics)" (p. 13).<br>"The performance agreements may usefully be viewed as the meeting place for public accountability and institutional autonomy appropriate to the particular sector, particular institution and particular region for two three year periods ( ... ). However, a legitimate question to pose would be: if there is some evidence of convergence of sectoral role and function, should not there be some convergence in the norms and character of resource provision i.e., the same payment for the same tasks—or, parity of esteem and parity of treatment. We would recommend this is kept in mind as matters evolve, as again, this is a phenomenon typical of most binary systems" (p. 82). +<br>"The development of tertiary education in Finland has progressed purposively over the last decade, both in polytechnic and university sectors. However, the conceptualisation of a university as an essentially Humboldtian construct, with strenuous career requirements; long courses of study; entrenched silo-like disciplines; and a limited managerial and steering capacity has clearly encountered difficulties, largely because of external imperatives. The evolution from a Humboldtian model to one of a modern university, with multiple objectives, diversified funding, purposive steering mechanisms and a strong external responsiveness is thus, in our opinion inevitable, and Finnish universities in different degrees display many of these characteristics" (p. 114). | **Lack of External Stakeholders:** "This absence of external stakeholders severely limits the institutions in their interaction with the external world, whose needs they are meant to serve. The effective formal separation of these two worlds, which are so mutually dependent in the knowledge society, is difficult to understand" (p. 65). +<br>**Lack of leadership:** "There is a perception, generally, that the leadership of institutions is weak. This is attributed in large measure to the particular method of selection of the rector, the totally internal focus and the rather political process involved. In addition, the structures within the university do not place a premium on the exercise of leadership. ( ... )" (p. 65). +<br>"Yet another issue is the excessive value placed on collegiality within the individual institutions. The many layers of decision-making and the large representative bodies that focus on collegiality ensure that processes are labyrinthine. The ineffectiveness of decision making arises from the multitude of statutory bodies and the excessively large size of these bodies.<br>The slowness of decision making, and the lack of clarity and transparency of decisions when they are made, seem to be an inevitable consequence of the structures and are totally out of step with modern and more effective approaches. Collegiality is a valuable concept in a higher education institution but we believe that this can be achieved within a framework of a reduced number of layers of decision taking, and with bodies with much smaller membership" (pp. 65–66) + "We were struck by the strong feelings expressed regarding the micromanagement of the system by the government. Examples included detailed control over new programmes and modifications to existing programmes. ( ... )" (p. 64). |

**Table A1.** *Cont.*

| Neoliberal Triangle | Finland (2009) | Portugal (2007) |
|---|---|---|
| Performativity | **Performance requirements and assessment:** "The use of internal research assessment exercises linked to institutional strategic planning and resource distribution is commended as a matter of course" (p. 47). "Quality assurance, of course, is closely connected with monitoring of performance in a strategic sense, which raises the issues of the robustness and adequacy of data-bases (KOTA and AMKOTA), and their use in accountability and even, resource distribution i.e., steering the system" (p. 78). "Our evaluation of the performance agreement process from an institutional perspective would be thus: ( . . . ) An appreciation of the lump-sum budgeting from government, in the realisation that management has to exercise judgement and a firm hand in subsequent internal distribution in a micropolitical competitive faculty environment" (pp. 82–83). "The need for really effective university training departments linked to strategic planning process and quality review. Universities should assess how their training departments should evolve, and what their new training priorities should be" (p. 91). **Performativity and Governing Bodies:** "In the event of the adoption of the "legal person" principle, it would be necessary to establish a governing body or board of trustees for each type of HEI accountable to government. ( . . . ) The purpose of such a body would be to operate at a strategic level, interacting with stakeholders, improving the institutional infrastructure, but not interfering in institutional management or the academic domain" (pp. 108–109). | **Performance:** "The section on economic performance points out that the level of human capital formation in Portugal is no longer able to sustain productivity growth levels that are needed to close the income gaps of the country compared to its competitors. The poor economic performance of recent years can be linked to the country's weak performance in human capital formation" (p. 19). + "The governing authority should be reluctant to establish additional statutory bodies and the creation of such bodies should be subject to the most rigorous examination and justification. Decision making should aim to be efficient, effective and transparent." (p. 69). **Accountability and Governing Bodies:** "A suggestion has been made to us that there should be a regional council formed in each region comprising all of the HEIs and other educational and training providers together with a broad representation of stakeholders e.g., from business, trade unions, voluntary groups, etc. Such bodies would not have a statutory or a decision-making base but would be a vehicle for local joint initiatives, for example. They could also have a role in recommending to CCES the realignment or the formation of new relationships among HEIs in the region. Annual reports on activities would be provided to CCES as an input to its national overview and its annual contract discussions with individual institutions. We believe that there is much merit in this idea and we suggest that government provide the necessary start-up funding to support the administration of such bodies" (p. 69). **Accountability and Performance:** "Revise budgeting for capital outlay: The review team recommends moving away from the project-funding mode for capital outlay, toward a multi-year plan for capital improvements, linked to national priorities. The plan should include attention to revenue sources for capital outlay, and anticipate the eventual loss of European Structural Funds. The criteria for capital priorities need not be identical to programme priorities, and can include factors such as regional economic growth, jobs, the preservation of buildings and sites of historic and cultural significance, and contributions to the civil society through the arts or service to communities" (p. 128). |

Table A2. How Portugal and Finland translated the OECD proposes based in three main components of neoliberalism—market, management and performativity (Ball 2016).

| Neoliberal Triangle | *Yliopistolaki* (University Law) 558/2009, Finland | Law 62/2007 (RJIES), Portugal |
|---|---|---|
| Market | The New Universities Act established universities as independent legal personalities, being either public universities or foundation universities.<br>"Section 5. Legal capacity of public universities: 1. The public universities are independent legal persons."<br>Following the OECD recommendations, the following institutions become foundations: "Aalto University Foundation operating as Aalto University, and TTY Foundation operating as Tampere University of Technology".<br>*New relation with the government:* "Section 5. Legal capacity of public universities: 1. The public universities may undertake commitments, obtain rights in their own name and possess movable and immovable property. The universities may engage in business activities, provided such activities support the discharge of the mission laid down in Section 2.<br>2. The public universities are liable for their commitments with their own funds and are entitled to pursue and defend litigation in court."<br>*Regarding tuition fees for international students and the typical free nature of Finnish HE:*<br>"Section 8. Tuition free of charge and charges related to other activities (Amendment 414/2016)<br>1. Studies leading to a university degree and entrance examinations relating to student admissions are free of charge for the student unless otherwise provided in this Act. (...)<br>"Section 9. Commissioned education (Amendment 1600/2015)<br>2. Tuition provided in the form of commissioned education must relate to undergraduate or postgraduate education in which the university has the right to confer degrees. ( … ) +<br>"Section 10. Fee-charging degree programmes: 1. Universities must charge a minimum tuition fee of EUR 1500 per academic year for students admitted to a Bachelor's or Master's degree programme taught in a language other than Finnish or Swedish. ( … )"<br>*Regarding the diversification of the funding basis:* "Section 49. Criteria for the allocation of government funding: 1. The Ministry of Education and Culture grants imputed core funding to the universities, taking into account the extent, quality and effectiveness of the operations and other education and science policy objectives. The Ministry of Education and Culture may also grant performance-based funding to universities on the basis of good performance. ( … )" | The new legal framework for HEIs (RJIES) created the possibility for HEIs to become foundations, a kind of ideal model in line with the OECD's proposal.<br>"Article 9—Status and legal framework: 1. Public higher education institutions are collected persons governed by public law which may, however, assume the status of public foundations governed by private law under the terms stipulated in Section III Chapter VI ( … )".<br>"Article 129—**The creation of a foundation.** (...) "The transformation of an institution into a foundation governed by private law should be justified on the basis of the advantages of adopting this managerial model and legal framework for the pursuit of its objectives".<br>Following the OECD recommendations and the legislation, three universities immediately assumed the foundational status: the University of Aveiro, Porto and ISCTE.<br>"Article 134—Legal framework: 1—Foundations are governed by private law, specifically with regard to their financial assets and staff management, apart from the exceptions established in the previous points".<br>*Redefinition of the institutions' governance bodies:* "Article 77—Governing bodies of universities and university institutes:<br>1—Universities and university institutes are governed by the following bodies: (a) *The General Council;* (b) The Rector; (c) The Management Board.<br>2—With the aim of ensuring cohesion within the university and the involvement of all organisational units in its management, the statutes may provide for the creation of an academic Senate consisting of representatives of the organisational units, which acts as an obligatory advisory body to the Rector on matters defined in the institution's own statutes. ( … )"<br>*Regarding Tuition Fees:* "Article 139—Tuition fees and other chargesTuition fees and other charges payable by students for attending educational establishments are established by the founding body on the recommendation of the managerial bodies of the establishment and must be announced and suitably publicized in all respects before students enrol".<br>*Regarding the diversification of the funding basis:* "Article 115—Income<br>1—The income of public HEI consists of the following: (a) Budget allocations received from the State; (b) Revenue from tuition fees and other charges for attending study cycles and other training courses; (c) Revenue from research and development activities; (d) Revenue from intellectual property; (e) Revenue from the institution's own assets or assets from which they benefit; (f) Revenue from services rendered, the issuing of expert opinions, the sale of publications and other products from their activities; (g) Subsidies, grants, partnerships, donations, inheritances and bequests; (h) The proceeds from the sale or leasing of tangible assets, or other assets, authorised by law; (i) Interest on deposit accounts and remuneration from other financial applications; (j) The revenue and expenditure account balance from previous years; (l) The proceeds from charges, salaries, fines, penalties and any other income to which they are legally entitled: (m) The proceeds of agreed loans; (n) Revenue from pluriannual loans agreed with the State ( … )". |

Table A2. *Cont.*

| Neoliberal Triangle | *Yliopistolaki* **(University Law) 558/2009, Finland** | **Law 62/2007 (RJIES), Portugal** |
|---|---|---|
| **Management** | *Removal of the civil service designation from all employees of universities:* "Section 32. Staff employment relations: 1. The employment relationship of the university staff is based on a contract of employment. 2. The employees and the terms of the employment relationships are governed by relevant statutes and terms agreed in a collective agreement and in the contract of employment. The universities will be able to pursue independent human resources policies, improve their attractiveness as an employer and in this way strengthen their competitive advantage in order to recruit the best personnel". | *Increased autonomy in terms of human resource management:* "Article 134—Legal framework—( . . . ) the institution may create career structures for its own teaching, research and other staff which, in general, parallel the teaching and research staff categories and qualifications of the various public higher education establishments". **Top-down management:** Inclusion of external members in the institution governance bodies (e.g., General Council (§81º) governing bodies) and strengthened leadership. |
| **Performativity** | **Evaluation and assessment of HEIs:** "Article 29—Registers and transparency: The supervising ministry organises and keeps an up-to-date official register accessible to the public of the following data on HEI and their activities: (a) HEI and their relevant characteristics; (b) Consortiums of HEI; (c) Current study cycles leading to the award of degrees and, where appropriate, the regulated professions for which the degree holders qualify; (d) Teaching staff and researchers; (e) The results of the accreditation and assessment of HEIs and their study cycles; (f) Statistical information, specifically on the number of places, applicants, students enrolled, degrees and diplomas awarded, teaching staff, researchers, other staff, student social services and public funding; (g) Employability of holders of degrees; (h) The general base of higher education graduates; (i) Other relevant data, as defined in an order issued by the supervising minister". "Section 51. Supervision and reporting (Amendment 954/2011): When requested by the Ministry of Education and Culture, each university must provide the Ministry with the data necessary for the evaluation, development, statistics and other supervision and steering of education and research in the manner determined by the Ministry. (Amendment 954/2011)" | **Assessment and accreditation of HEIs:** "Article 147—Assessment and accreditation of HEIs: 1—Under the terms of their statutes, higher education institutions must establish mechanisms for regular self-evaluation of their performance." + "Article 148—Supervision: HEIs are subject to the supervisory powers of the State and must collaborate faithfully and promptly with the appropriate authorities". |

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
