# Peer review of "Brothers in Arms? How Neoliberalism Connects North and South Higher Education: Finland and Portugal in Perspective"

_socsci, doi:10.3390/socsci11050213_

Round 1

Reviewer 1 Report

The paper presents and discusses an original and relevant research. Exploring how intergovernamental organizations frame national policies and promote convergence despite each country specificities is a quite stimulating endeavour. It would be interesting to add the authors' views on whether the results would be different if considering countries with different characteristics (not small and peripherical, with different HE systems,...). The criteria and procedures that are on the basis of the content analysis of OECD reports are not sufficiently detailed in the paper. There seems to be a mistake in the title of section 5.2 (line 430): it is written "market" but it is probably "performativity" or "performance"

Author Response

The authors are very thankful for the comments provided by both reviewers, which helped to improve the overall quality of the paper. The authors have considered them seriously and the article went through some revisions. Most of the changes and editing was carried out through track changes, so that the reviewers could assess the transformation of the article. Additionally, separated comments to reviewers’ feedback is provided:

Thank you for your comments.The authors’ views on the fact that results would be different if different countries would be considered with different characteristics would have been considered for analysis were addressed, but mostly in the conclusions section. However, the authors ground on the literature, namely on Ong (2006) in an attempt to signal this comment raised by the reviewer (at the end of page 2, lines 92-95): “In fact, Ong (2006) looks at neoliberalism as a flexible “technology of governing” used (enthusiastically and differently by different countries and regimes, according to their political administrative traditions (be they authoritarian, democratic, or communist) in order to compete in the global economy.” 

The criteria and procedures that are on the basis of the content analysis of OECD reports were further clarified in the methods section added in conclusions.

This was a mistake, thank you for highlighting it. We have corrected it.

Reviewer 2 Report

The article addresses a very interesting topic and presents the results clearly. The purpose of the study is described in a logical, comprehensible, and explicit manner.

However, the authors should also consider the following suggestions:

  1. Introduction

-Line 27: The abbreviation HE seems to be repeated.

  1. Materials and Methods

- Line 323: Although Table A2 is mentioned several times in the text of the paper, it does not exist in the Appendix section.

  1. 2 The Market

-Line 430: Title 5.2 The Market I think is wrong. It seems that this section is about the Performativity category.

  1. The versatile normative side of the OECD: Convergence in translating neoliberal 569 recommendations

- Although the authors highlight the conclusions of the study, they do not present the implications of this paper. In this regard, I think that the author (s) should better highlight the implications of their findings on all stakeholders (HEIs, researchers, others). Also, perhaps the contributions (implications of this study) should be grouped into theoretical and practical contributions.

- The authors did not present the limitations of the research. I propose to the author (s) to address these elements as well, if they have been identified.

Appendix

- Appendix A: Although the authors mention that Table A1 refers to the three main components of neoliberalism, only the Market component is shown in this Table.

Author Response

The authors are very thankful for the comments provided by both reviewers, which helped to improve the overall quality of the paper. The authors have considered them seriously and the article went through some revisions. Most of the changes and editing was carried out through track changes, so that the reviewers could assess the transformation of the article. Additionally, separated comments to reviewers’ feedback is provided:

Thank you for your feedback.

This was a mistake, thank you for highlighting it. We have corrected it.

There was a problem with the formatting of the table – we hope now, in pdf format, it is visible.

This was a mistake, thank you for highlighting it. We have corrected it.

The implications of this study have been better elaborated in the conclusion section (The versatile normative side of the OECD: Convergence in translating neoliberal 569 recommendations), as well as its limitations that can actually configure as further research. We have limited other research to the previous OECD reports, and the latest reports will be object of further analysis to track continuity on these trends. Thank you very much for bringing this point out.

There was a problem with the formatting of the table – we hope now, in pdf format, it is visible.

Sara Diogo and Teresa Carvalho.
